# Holding Our Nerves—Experiments in Dispersed Collective Silence, Waking Sleep and Autotheoretical Confession

**Grace Denton**

Department of Arts, Northumbria University, Newcastle upon Tyne NE7 7XA, UK;
grace.e.m.denton@northumbria.ac.uk

**Abstract:** As part of my practice-based research, I host a monthly radio show based on the principle of 'waking sleep', resulting in a largely silent experiment in dispersed communion with an audience. Silence—though frowned upon in standard broadcasting—has long been a feature of artworks from Marina Abramović (1973–present), to John Cage's *4′33* (1952), to Gillian Wearing's *Sixty Minutes Silence* (1996). The power of collective silence is harnessed by many doctrines: in Quaker meetings for worship, in the practice of Zen Buddhism, and in the Memorial observance of a minute's silence. The practice of 'waking sleep' was coined by Ned Hallowell M.D. as a means of refreshing the brain and combatting the effects of ADHD. It is simply the act of letting the mind wander, without feeding it the next dopamine hit from a stimulant like a conversation or screen-scroll. Holding My Nerve is a radio show, and an ongoing autotheoretical artwork. It is part-field recording, part-endurance performance, and tracks my research process as it evolves. Using transcripts of the show, diaristic writing, and reflections on art history and my past works, this article explores the often-fraught relationships between autotheory, visual art, neurodivergence, and practice-based research.

**Keywords:** silence; performance art; autotheory; neurodivergence; ADHD; practice-based research





## 1. Introduction

*I think it's really important to know that all of the so-called theory of philosophy, comes out of the urgencies felt by the human being, being alive. A lot of the liberatory theories: queer theory and feminist theory and critical race theory, they're about . . . they couldn't be more about life.* (Nelson 2015)

—Maggie Nelson

Autotheory, as I see it, is about the re-wedding of theory to the everyday, to the very stuff of life. In working autotheoretically we can inflect theory with the personal, grounding philosophical debate amongst what Berlant calls "the reproduction of ordinary life" (Berlant 2011, p. 98).

This act has a multi-pronged effect on the artist. Rather than simply rolling out confessions in a cathartic stream, autotheory takes the self apart. To paraphrase and gather the many terms used by Lynne Huffer (Huffer 2020) to describe the autotheoretical process, the artist is self-unsaying; self-undoing; self-unbinding; self-hollowing; self-writing; self-registering; self-bracketing; self-interrupting; rendering the self on fire; self-blinding; self-inventing; puffed up self-regarding; self-slamming; self-annihilating; self-piercing; and tacking scraps of [the] self to the bulletin board. From these multiple, often opposing truths that emerge, something resembling the complexity of a lived experience can be observed. By parsing personal fragments alongside philosophical, critical theory, we uncover something rich and nuanced.

Although popularised by a recent lineage of writers and academics (Stacey Young, Lauren Fournier, Maggie Nelson, Paul Preciado), I argue that autotheory has been practiced by visual and performance artists for decades. I accompany this reframing of past works with reflections on my own autotheoretical practice, assisted by this new language, and

critical development in art and literary history. Fournier defines autotheory as taking "one's embodied experiences as a primary text or raw material through which to theorize, process, and reiterate theory" (Fournier 2018). From a practice-based point of view, this analogy works particularly well, as artists can make use of their actions and physical contexts as a medium. In this article, I expand upon my radio show *Holding My Nerve* (Denton 2021–2022): one output of my current practice-based research. In doing so it becomes pertinent to reflect on previous works, to hold and digest them, making tangible what was a largely intuitive and ephemeral progression, and re-framing it alongside this new autotheoretical (anti)canon.

## 2. Self-Reference and Performing Assimilation

In the conclusion of Lauren Fournier's recent pivotal text, she describes the 'self-immolation' (Fournier 2021b, p. 267) required to fit into the intellectual confines of the academy, self-immolation being the 'deliberate and willing sacrifice of oneself, often by fire' (Merriam Webster Dictionary n.d.). This language evokes violent images: immolation as pointed spectacle, performed for a cause; an ending or destruction of one's physical form. The self-immolation of the institutional environment is violent, but it is a violence of an insidiously creeping, quiet nature. We immolate by a thousand tiny cuts, encouraged to erase our subjective reality in favour of a neutral voice. We become "agreeably opaque" (Fournier 2021a, p. 268).

However, the issue with describing the institute as a place of self-immolation, is that this compulsion is not universal. There are those who do not feel the need to erase themselves, who are already central to this monolith, unwittingly reinforcing their supremacy by neglecting to name themselves. This phenomenon was acknowledged by Roland Barthes in 1957, in relation to the bourgeoisie. I first learnt this term in my partner Craig Pollard's thesis, where he tied Barthes' definition of the term exnomination to whiteness.

> [ . . . ] it is worth turning to critical understandings of *whiteness* (as explored by scholars such as Sara Ahmed) and also the notion of *exnomination* [ . . . ] a word credited to Roland Barthes, literally meaning 'outside of naming'. Barthes uses the term in 'Myth Today' when describing the different ways *bourgeoisie* (as a concept) operates in relation to 'economic', 'political' and 'ideological' discourses, [ . . . ] in *not* referring to itself, the bourgeoisie and its values are (insidiously) able to become naturalised and accepted as an ideological societal norm. And this phenomenon has been echoed by writers and critics when pointing to a similar process of un-naming that occurs regarding the centrality of white-centric socio-cultural discourse in the West. [ . . . ] societal standards become ideologically informed by a white perspective which mediates all subsequent human experience; whiteness becomes the standard from which all disparate meanings diverge, and "difference" and "otherness" are constructed and understood primarily in relation to *white*. (Pollard 2018, p. 18)

The popular image of the scholar is still a relaxed white male, of any age but perhaps preferably advancing in years, brow furrowed, surrounded by tottering piles of books, slightly scatty about the everyday mechanics of life, but he can be forgiven! Because he's thinking! He probably has someone else to make the wheels turn, or he's comfortable in chaos. Although this image is vividly (and cynically) painted, I would argue that it persists. These specificities are reproduced, and absorbed as neutral, into the reality of the institute. I see it in the workload delineation of University departments, in the slow migration of the weight of responsibility towards female and otherwise underrepresented staff. The concerns, research and influence of these white males are undisputed pillars of the ecology, immoveable until they finally retire out of the system or climb to a higher pay grade elsewhere. Paying witness to this dynamic is an exhausting reminder: the specificity of anyone outside these pillars is seen as inconvenient, unseemly, and best erased. It is a truth communicated in looks and omission, in a lone question mark on a manuscript, never elaborated.

Perhaps in defiance of this context, the methodologies of my research have developed as obtuse yet quiet refusals. A radio show punctuated with silence and frowned-upon confessions. Autotheory likewise has largely been framed as a feminist practice, addressing historical omissions by inserting the self forcefully into the canon. It is "the influential men of art and academia who occupy the realm of the mind, and the women who always already occupy the realm of the body" (Fournier 2021a, p. 234).

In 2015 I began my MFA, following a course I had set myself into the Art World Proper. This new space felt alien, and I observed myself learning a new language (visual and literal), acclimatising to the environments (the staged-feeling studio) and choreographing my body (the stances and poses of the group crit). My previous artworks had been outward facing, I pointed my lens towards others, documenting my chaotic life through a 35 mm lens, and then clumsily attempting a collaborative film with my Nanna. By the time I reached my graduate studies, I had become paralysed by the thorny ethics of Documentary, of testing out my ideas and fledgling craft on people close to me. Through necessity and curiosity, I turned the focus onto myself, and began to perform my imposter syndrome. My self-referential videos and performances exaggerated my own learning process, clowning my assimilation. I also unwittingly acknowledged the difficulty of navigating academia first time round, with undiagnosed ADHD and fresh out of school, when I had undertaken a Bachelors course at the University of East Anglia. As an ambitious yet unprepared teenager, I had left the relative structure of the family, and entered self-guided study with a disorientating bang.

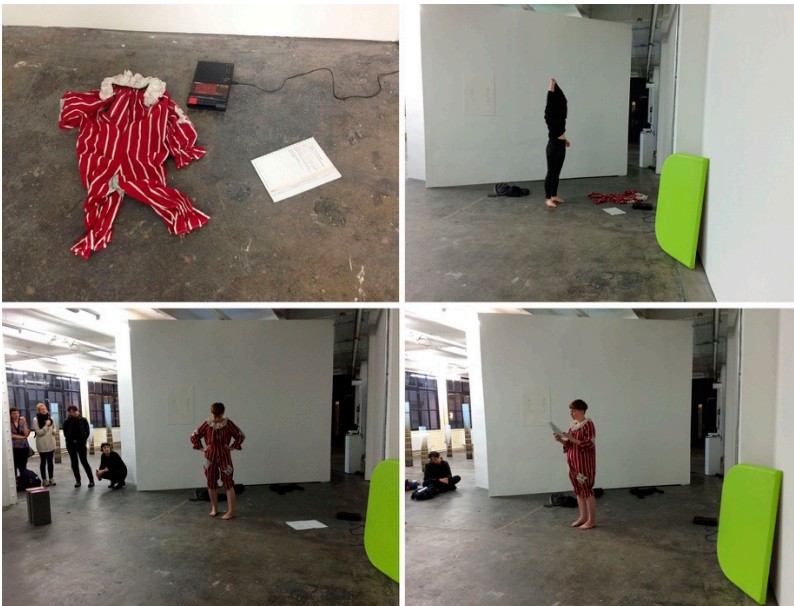

**Figure 1.** Denton, Grace. *Sixty Five*, performance with tape player, clown suit, and BA dissertation Part of *There Were Islands* BxNU MFA performance event, BALTIC 39. June 2016. Images courtesy Craig Pollard.

I began the performance of *Sixty Five* (Figure 1) by changing into a clown suit, as my pre-recorded voice narrated a story for the audience:

> In 2009 I was at the end of a degree in English literature. I was writing a dissertation. It was the first time I'd felt positive about my writing. The course had so far taken away my ability to read, to absorb information, but this was slowly starting to return. In our final meeting, my dissertation supervisor asked me what marks I had been getting. My supervisor was the head of English. My supervisor was a very busy man. My supervisor was coolly interested. I told him I'd been averaging at a sixty five. But I was hopeful that I could push that to a seventy with this paper. I had been getting better. My supervisor made a note. When

my bound dissertation was posted back to my family home in the summer, my dad unwrapped it. Over the phone he told me there were no pencil marks on the pages, no notes on the cover sheet, it didn't look like it had been opened at all. The mark I received was sixty five. I've not read a word of it since then. This is the first time I have opened the pages . . . [1]

This piece in particular became a quiet accusation squared at my past self, and my past supervisor, exploring the line between self-flagellation and self-protection by forcing myself to look back, to open a painful wound sealed shut by apparent neglect.

Between 2009 and 2015, I had worked in the arts, finding comfort in moving through different organisations, learning quickly, growing bored, realising that although I could easily organise and market artistic events, it was the actual making of the art I craved. I slowly course-corrected and made my way into art school, finally occupying rooms with paint-covered sinks. There they did not question *why* you were making something, they simply accepted your vocation, instead questioning the decisions within the piece itself, helping the maker to refine their vision, reflecting back what an artist communicates and offers to the world. I observed myself responding well to this new environment, being able to use my age and experience to greater effect this time, still anxious and occasionally paralysed—but this structure, this rigour, these personable encounters with tutors who were practicing artists themselves, all contributed to keep me on track.

In *The Year of the Clown* (Figure 2), I performed in the glare of a blank projector image in a parody of an artist talk, speaking from notecard prompts that contained just one word I improvised on—"the work is about *flatness*, and the implicit 2D nature of our relationship to the replicated world, and the planes of vision we use to inhabit the"—etc., etc. These empty phrases poked affectionate fun at the world I so desperately wanted to be part of, feeling at once drawn into and repelled by its conventions. The block colour 'slides' changed of their own accord, referencing traumatic PechaKucha[2] presentations of my past, and my horror of public speaking. I wrote lists, litanies of the implicit rules I felt a performance artist was held to, I printed them on banners and hung them in student shows. I was performing my own inadequacy, transferring brattish defiance into artistic form.

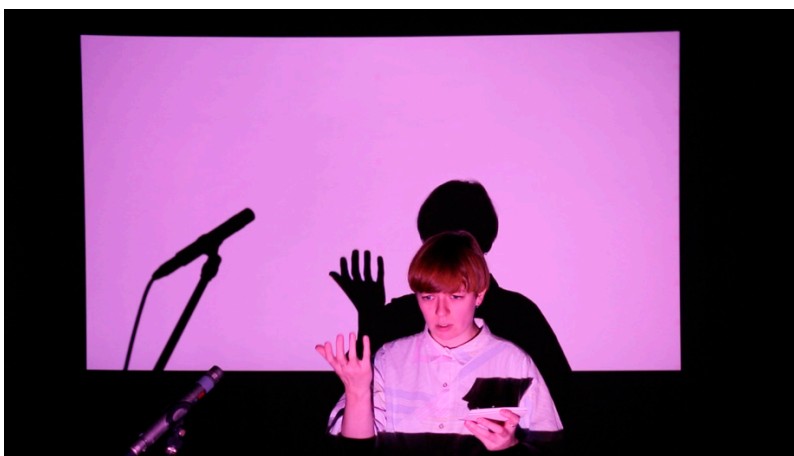

**Figure 2.** Denton, Grace. *The Year of the Clown*, performance with digital projector and note cards. Part of *Playing Myself*, in response to *The Bodyssey Odyssey* by Pester & Rossi, BALTIC 39, December 2016.

An emerging pattern in my work became the gentle refusal of audience expectation and satisfaction. I eschewed the abrasive and aggressive aspects I had presumed were par for the course in performance art, and instead made a quiet space of absence where the 'content' would usually be. *In The Telling* (Figure 3) involved a silent reading from a piece of hand-written text, I mouthed the words, and only myself and Craig knew what was written there. I surprised myself by emotionally responding to the text and beginning to cry mid-performance. I had heightened my own (and the audience's) responses by playing

a slowed down version of a Roxy Music song on a tape player next to me and stopped the tape when I had finished reading. Or when the song ended? I forget.

Now, this refusal is demonstrated in the silence of my radio show, and the constant push and pull of embarrassment I feel about taking up the airwaves with 'nothing' but my own dogged determination. This silence and refusal is arguably the opposite of the ADHD impulse, with its need for constant occupation, but it also feels like the wall put up by my brain and by my condition. An impenetrable forcefield between myself and the things I want and need to do. It is the way my attention slides off things imperceptibly. It is the invisible barriers that can somehow evaporate at unpredictable moments, and seem completely perverse to others.

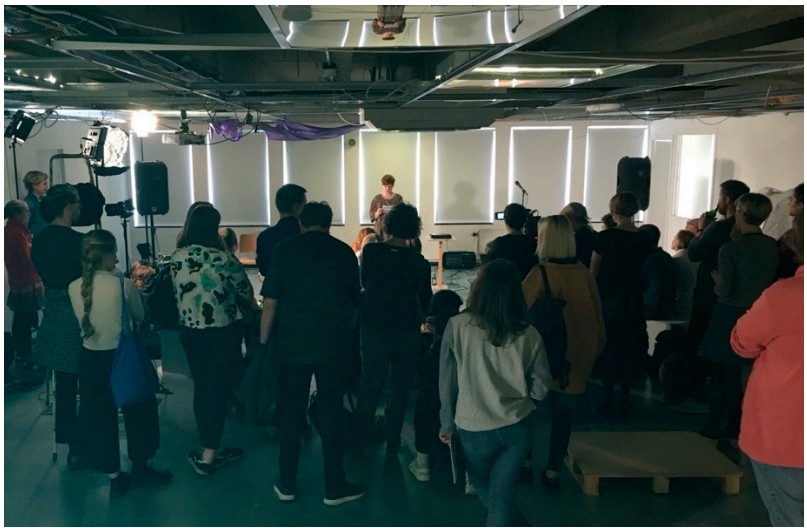

**Figure 3.** *In The Telling*, performance with tape player and handwritten text Part of *film (Studio is Sudden)* by Kathryn Elkin and Giles Bailey, Northern Charter, July 2017. Image courtesy Giles Bailey.

## 3. Holding Space for Silence

Holding our nerves requires gritting our teeth, persevering against all odds, maintaining a certain level of control. I am interested in the parallel act of holding onto control by surrendering yourself to a theory, and to the judgement of the unidentified, the unseen, the institute.

There are many theories, life hacks and simple fixes proposed to focus the wandering ADHD mind. Although unable to implement many of these 'fixes' into my life, I have a voracious appetite for reading about them. I heard about 'waking sleep' from a practitioner with personal experience of the condition—Ned Hallowell, M.D. The premise is deceptively simple.

> I allow time for daydreaming (without screens), thinking, or staring off into space. This is 'waking sleep', and science has shown it's actually good for us. Just a few minutes of waking sleep will give you a huge brain boost to get through the rest of your day! (ADDitude 2020)

Although I have yet to find a source for the scientific research he mentions, this idea became key to a performance I had imagined for some time. A guided meditation[3] with no guidance, a mutual quiet moment across radio waves; if a silence is shared by multiple people across fractured time and space, is it really shared? What effect would this repeated attempt have on my brain, and my practice? I already knew that my own neurodivergent mind responds well to silence. In the middle of the night, when all is quiet, a new kind of focus appears. Many of us had to guiltily admit that when the world slowed right down in March 2020, we finally felt at peace. I am trialling if it is possible to carve out that space for myself, amongst the (internal and external) noise.

Since June 2021, I have broadcast a monthly hour of radio (Figure 5) in which I attempt waking sleep, and encourage it amongst the listeners. Together we access a space defined less by a specific approach to wellness, more by our own needs, perhaps our own subconscious. There has been no more useful, no more difficult or more confounding experience than trying to sit with the self. Is the opposite of self-immolation . . . self-centering? This is an accusation regularly levelled at autotheory. But this action feels less like autobiography, or egoism, rather a shared acknowledgement of reality. I allow a theory on my particular type of brain to pass through me, to be inflected by myself.

In a medium in which audibility is a currency, the aim to remove, restrain and minimise sound is perverse. To a listener, any silence longer than a few seconds is unexpected, even unsettling. Why would you continue listening if there's nothing to listen *to*? The movement of the progress bar the only proof that something is being played.

In some part of my mind I believe that my radio show, which is pre-recorded, yet broadcast live, and then later archived, is a tiny quiet private area of the internet where I can say what I like and no one will *really* listen. I can whisper my most revealing confessions and they will be shared by a scant few who may need them. Although it is immortalised in the great hypernetworked archive, it feels protected by its absurdity. For those who listen, I hope it maintains that sense of privacy, that cloaked invitation.

Silence can have many folded meanings. There is a language of silence which means compliance or consent. There is a language of silence which is loaded with reverence. This silence I embrace is part communion, part endurance. I think of the silent practice of waking sleep as a physical embodiment of a theory—I am processing an idea, an approach, a nonmedical intervention, with my body. The act of sitting without stimulation is extremely difficult, particularly for my type of mindbody. Marina Abramovic apparently said "the hardest thing to do is to do something that is close to nothing, because it is demanding all of you" (Morgan 2018). This act of silence is an act of endurance, but as the shows progressed, I decided not to stifle myself to maintain absolute quiet. The performance became about simply existing, at times I had to own my inability to be still, and refrain from editing out the need to blow my nose, fidget, sigh, gasp. I am reclaiming silence, rather than maintaining it as pristine, sacrosanct; I am filling it with this act of communion. I trust that any external silence is understood as masking a churning inner world that this daydreaming allows space for.

Dr Devon Price recently had a breakthrough: mindfulness is surprisingly useful (Price 2022). He shifted away from seeing it as something that will calm or soothe us, and instead, argues that it simply allows us to exist as we are. As neurodivergent people, we avoid the present, because it is hell. We project into possible futures, escape into distractions, and even learn to embrace or enjoy doing so, rather than feel shame. "The turmoil I'm experiencing has always been here. I can't fix it. I can only accept it." (ibid.). Waking sleep is something I would not have been able to do without the structure the radio show has provided: without an obligation, an audience, a monthly deadline, without a reminder to experiment with this novel practice. Through repetition I slowly become comfortable with this concept, this state. As Price discovered when he allowed himself to remove mindfulness from a solutions-based context—we are simply trying to be *present*.

In Laurie Anderson's recent Norton Lecture *The Road* (Anderson 2021) she interrupted her frenetic-but-focused ruminations with—"Meanwhile, there are so many things that computers can't do well, and the most obvious thing they can't do, is this . . . . . . . . . " She fell into silence. She held her nerve. Her eyes moved occasionally to the camera, to the outside of the frame, to her hands on the controls of her live-mixing desk. Eventually she began to flash a message onto the screen. "There is nothing wrong with your internet connection" (Figure 4). Afterwards she observes: "there's no such thing as digital silence, either it's on, or it's off." The closest we get is a pause to buffer. Even the subtitles were confused by her act—the auto-captioned tail-end of her previously uttered sentence stayed visible until she spoke once more.

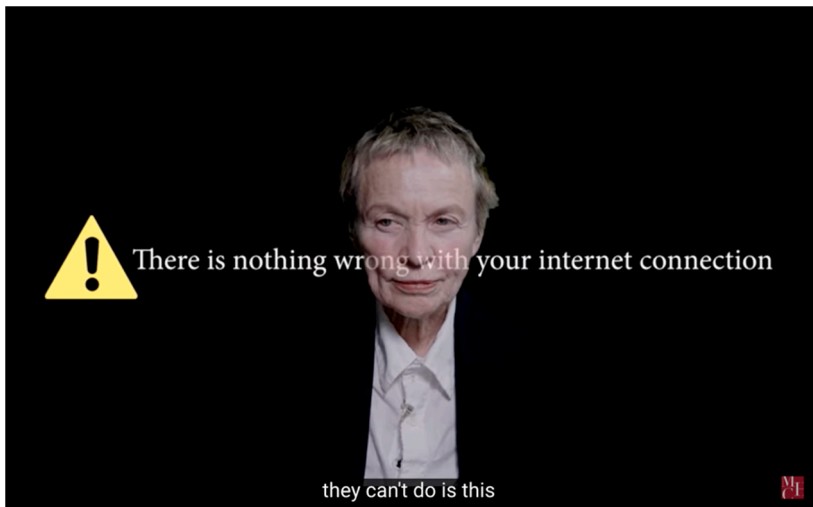

**Figure 4.** Anderson, Laurie. Still from *The Road* part 4 of Norton Lectures: Spending the War Without You. October 2021.

In order to observe my shows as a text, I put the audio files through a transcription AI. The results are not as garbled as you might think, and yet the fine detail is lost. The names and medical terms become odd sections of slang, or an incongruous brand name is thrown into the mix—the algorithm is always taught to serve capital. Like Anderson's auto-captioning, the AI cannot translate silence. The flow of words bunch together, with time stamps thrown in arbitrarily.

In 1952, the very first performance of Cage's 4′33″ took place at Maverick Concert Hall, with virtuoso pianist David Tudor. "Keeping track of time by consulting a stopwatch and reading a now lost, blank staff-notation score in 4/4 time, Tudor did not play a single note. Instead he sat in concentration, letting the listeners react to this musical non-event. The response, even from an audience accustomed to experimentation, was not positive. Many felt 4′33″ was an arrogant joke, [ . . . ]. For Cage however, 4′33″ was absolutely serious" (Kamps et al. 2012, p. 63). The piece is of course indebted to deep-rooted traditions of the practice of silence, and in particular Zen Meditation[4]. However, as with so many celebrated male artists, we do not see the workings, and "Cage was unclear on whether his formal studies of Buddhism predate 4′33″ (Kamps et al. 2012, p. 64).

When I first began thinking about silence, I was almost reluctant to acknowledge the contribution of this work. It has become so ubiquitous as to induce an eye-roll, however reading the description of its debut performance left me thrilled. I felt the performance vividly, from decades away, having experienced this state of performative silence for myself. Cage described the two distinct bodily sounds which become audible in deep silence, a phenomenon I've also shared—"the low tones of his blood's circulation, and the high frequency buzz of his nervous system" (ibid.).

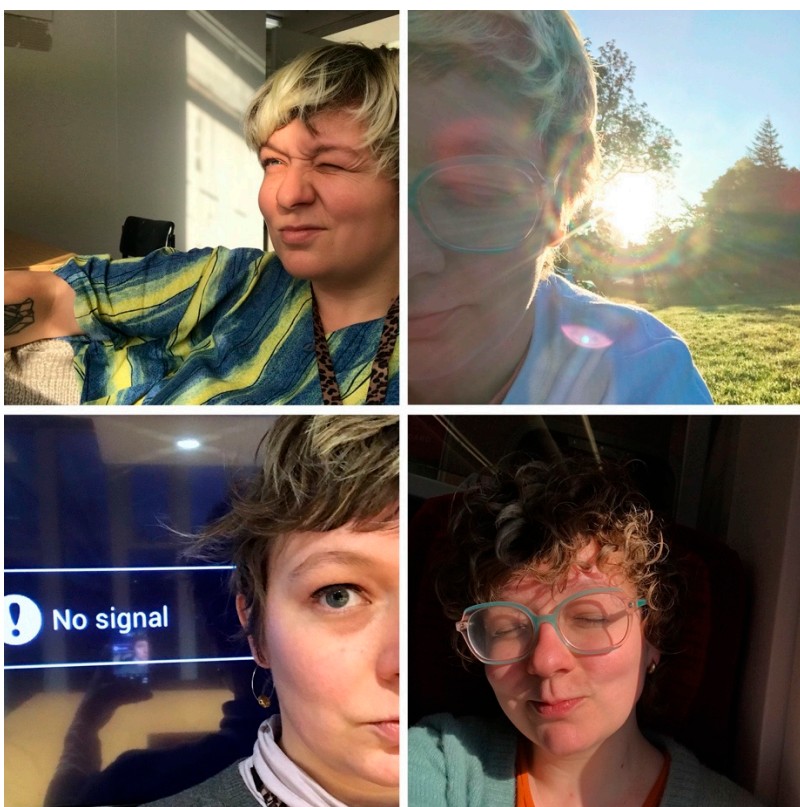

**Figure 5.** Tracking the passage of time: a prompt from the fledgling community radio station results in a new 'algorithm-friendly' selfie to accompany each show. Image icons for episodes 4, 6, 8 and 11 of *Holding My Nerve* Grace Denton and Slacks Radio, 2021–2022.

### 4. Extract of Episode 1 of *Holding My Nerve* as Transcribed by Otter.ai Grace Denton and Slacks Radio, June 2021

00:04

*Morning. Hi, this is Grace. So the basic aim of this show is that you and I do nothing for the next hour*

00:31

*sounds good, right? So each week, I will be in a different space or scenario. And I'll record for an hour. So that might be in a park, or library, any kind of public space where I'm unlikely to be disturbed, I will sit and not have anything to occupy me, like scrolling my phone, or listening to a podcast, or all of the things that I usually use as a crutch. And it would be nice if you want to join me and try and hold your nerve and do as little as possible. So the basic idea is the Our minds are very busy. And as easy at the minute for anxiety to creep in. In this capitalist, neoliberal hellscape, in which we live, feels like productivity is key. And aren't to try and gently resist that, I guess. For myself, as well, I have ADHD, which means I have a naturally wandering mind. And I feel like I always have to have multiple things on the go.*

03:00

*Or they'll kind of get bored or spin out or any number of bad things. So even though doing nothing is so good for us. It's very hard to implement. Because there's always there's always something we could or should or want to be doing. I recently found out through my research and interest in I don't have a complicated interest in mindfulness and meditation. Kind of weary, weary interest in those things, because I struggle with the language and the kind of implication that we have to solve our own problems when actually the structures that we live in make that impossible. But as we do exist in these structures, and it's incredibly difficult, even as a community to resist them. I thought this show and this idea might provide some solace are some feeling of being together or achieving*

*something together. So I recently found out about this well, it's a it's a practice called waking sleep. And it's apparently incredibly refreshing for our brains and helps us to focus and increase the kind of connections within our brain and kind of go deeper rather than existing just on this sort of anxious surface level. And in order to achieve waking sleep, you literally just do nothing you daydream and allow your thoughts to take you where you want to go. And if you have any experience with meditation, then you'll know that the the main idea is is to kind of focus on the breath and ignore or push out any other thoughts, but I feel like that's very difficult, and perhaps sets you up for failure because yes, focusing on the physical act of breathing can help you to slow your racing mind. But if you're feeling guilty before you even start for myself, I know that I'll never stop my mind from ping ponging all over the damn place*

07:15
*so yeah, my challenge to myself every month when I do this show, will be to sit or lie in in silence with no nothing to do with my hands nothing to particularly look at except the passing world*

07:44
*and just in an unguided way be present and to be with you all the listeners collectively, even though we're we're out of time and space from one another*

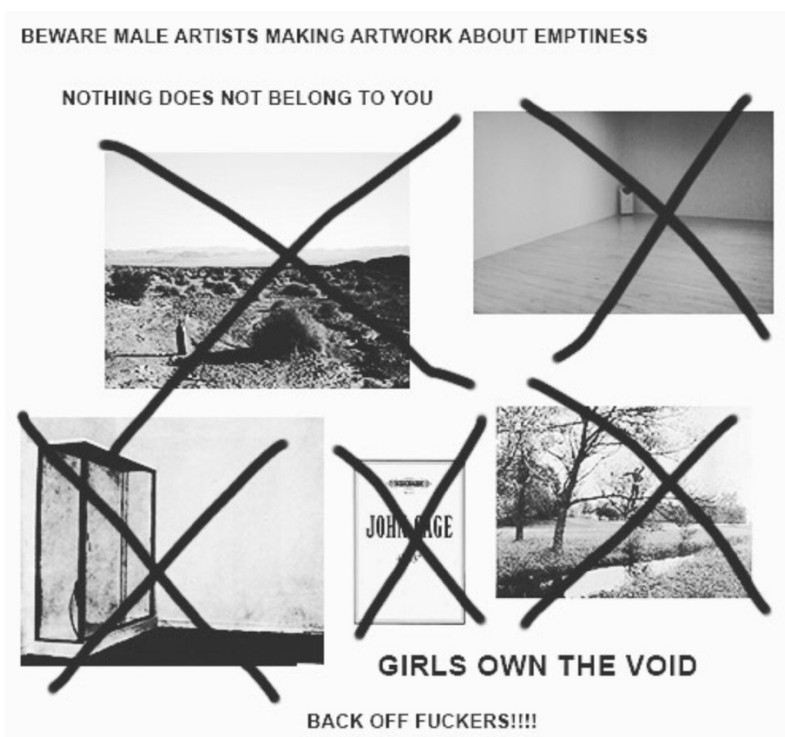

**Figure 6.** A PSA brought to you by ur local chapter of Female Nothingness, Audrey Woollen Instagram post, October 2015.

## 5. ADHD, Diagnosis, and Maintaining the Garbage

The epiphany of diagnosis can feel like an end in itself. Finally I have language to describe something previously so ineffable! Something that previously drew blank stares from those I tried to explain it to. An invisible force that stops you from doing things? Haven't you tried . . . just . . . doing things?

On this moment of realisation, Dr Gabor Maté wrote "It seemed to me that I had found the passage to those dark recesses of my mind from which chaos issues without warning, hurling thoughts, plans, emotions and intentions in all directions. I felt I had discovered what it was that had always kept me from attaining psychological integrity: wholeness, the reconciliation and joining together of the disharmonious fragments of

my mind" ([Maté 2019](), chp. 1). However, Maté acknowledges it is "both exhilarating and painful" to be able to give a name to your affliction. It is now possible to pinpoint the exact cause of your inability to self-govern, and it is great news (the possibility of help and salvation) and exquisitely sad (what-might-have-been).

Self-recognition can thankfully be powerful enough, as medical diagnosis is an obstacle course of administrative pitfalls and miscommunication (Figure [7]()). Aligning yourself alongside or against a neurotypical scale is a degrading process. The language of the medical establishment does not help. The second D in ADHD stands for 'disorder', but when spoken directly, the word feels like a slur. The term, as well as the common understanding of the condition, is outdated.[5]

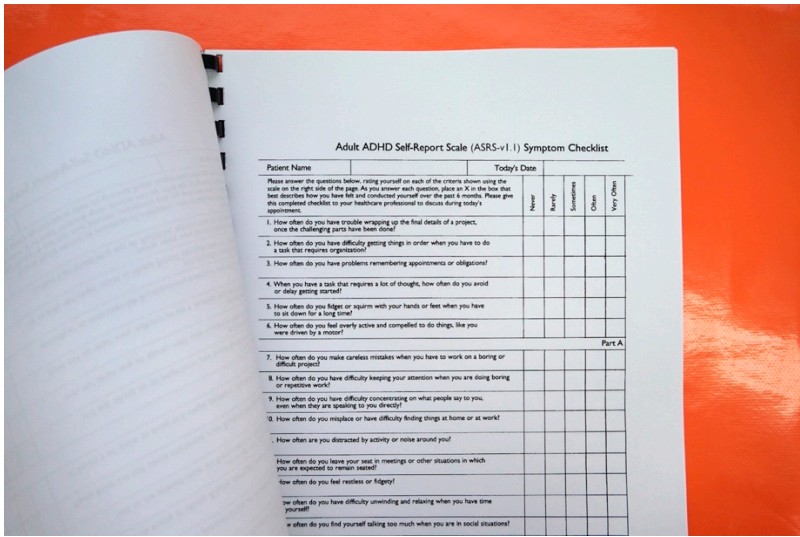

**Figure 7.** The standard diagnostic form used by the NHS to determine whether a patient requires a referral for Psychiatric assessment, which is also deployed before each subsequent visit after diagnosis.

ADHD is described not by the internal experience of the person suffering, but by the effect of their disabilities on those around them. In actuality, a sufferer of ADHD experiences an *overabundance* of attention, rather than the accepted *deficit*, it is a proliferation of thoughts that is difficult to stem. This often means the language around the disorder is impossible to identify with, and may never be diagnosed as the cause—for instance when *stimulant* medication is suggested, the sufferer is likely to recoil in horror at the idea of greater brain activity, and instead self-medicate with a relaxant such as marijuana. ADHD is also often misdiagnosed as anxiety, which I would argue is actually a symptom brought on by the inability to engage in the everyday.

Johanna Hedva describes this disconnect between language and experience in *Sick Woman Theory*: "It's important that I also share the Western medical terminology that's been attached to me—whether I like it or not, it can provide a common vocabulary: 'This is the oppressor's language,' Adrienne Rich wrote in 1971, 'yet I need it to talk to you.' ([Rich 1971–1972]()). But let me offer another language, too. In the Native American Cree language, the possessive noun and verb of a sentence are structured differently than in English. In Cree, one does not say, 'I am sick.' Instead, one says, 'The sickness has come to me.' I love that and want to honor it" ([Hedva 2019]()).

I am interested in framing this navigation of the oppressor (or the institution)'s language, as autotheoretical. This experience of living with a diagnosis, making work in response to and in collaboration with your own neurodivergence, means you are in constant dialogue with the theories, definitions, limitations and whims of the medical establishment. In order to receive the support you need, you must learn the language of the removed. The autotheoretical approach of researching and making while neurodivergent is an ongoing navigation of, and a reluctant/exuberant collaboration with your own neural idiosyncrasies.

We can trick our brains into allowing us to do the work, by doing the opposite of masking, by making autheoretical confessions, by embracing and *going with*.

Fellow researcher Jo Hauge[6] has developed a persona they adopt called Boy Witch, and is ALL ADHD: pure id, pure drive, pure ecstatic menace. Jo performs as Boy Witch while researching. Boy Witch does not want to do a PhD, he wants to write, a lot, but he also wants to eat the paper at the end.

I am interested in the conditions for working, so nebulous and shifting, the ideal environment to focus is always just out of reach. So much of ADHD-researching-making is focused on preparing these conditions: the maintenance of the basics to hopefully rise above them. In continually pushing at my limitations to do this, I am reminded of Maintenance Art, practiced by Mierle Laderman Ukeles when her artist status was dismissed after becoming a mother. In cleaning the steps of the institution, in collaborating with the Sanitation Department of New York City, Ukeles succinctly levelled the divide between the scholarly artist, and the lowly worker. "The sourball of every revolution. After the revolution, who's going to pick up the garbage on Monday morning?" (Ukeles 1969). Could this incessant self-maintenance, this bargaining and collaborating with our own neurodivergence, be seen as a new leg in the trajectory of Maintenance Art? We cannot phase out, ignore or mechanise this toil—maybe it *is* the work, maybe making that visible could be the truly radical, creative act.

When working autotheoretically we risk being dismissed by the Institute, by the impersonal academic, by the quota of thinkers for whom this is all thought, this is all academic, this is all held at arm's length from the body. We risk bringing the brain and the body together. We embrace the personal, we recapture our voices and ignore the haggard conventions. We acknowledge our subjectivity, because we know there is no neutrality, because we've repeatedly found ourselves outside of and at odds with that assumed middle ground. We cannibalise ourselves, our lives, and although we see it as raw material, others may not read it in the same manner. We risk being tarred by another brush, having our confessions misquoted, read out of context, used as proof of a different thesis. Returning to Lynne Huffer, she admits "there is something monstrous about this way of imagining my autocollage, seeing myself as detritus for a literary montage of garbage-writing. However, the monstrosity is not self-deprecating: it's not personal. Just being in what is." (Huffer 2020, p. 168).

### 6. Extract of Episode 2 of *Holding My Nerve* as Transcribed by Otter.ai Grace Denton and Slacks Radio, July 2021

51:19

*Mostly I was thinking about this thing about meditation. And I think my negative experiences of it have been when the focus has been on clearing the mind completely, and on pushing out any thoughts that drift up? And, yeah, I don't find that particularly useful, because I don't think I'm ever going to be able to do that. And I think a lot of people maybe feel similar. But I think I think it's connected to the fact that a lot of Western interpretations of meditation maybe focus on discipline, and its connection to productivity and kind of optimising the self. And I think, I think going into the practice of meditation. I always feel like if I'm not getting up at dawn, and clearing my mind then I'm already failing and I think for someone who struggles with routine and*

54:25

*just the basics of life getting up at the same time, every day is impossible for me. So enforcing a habit of something as difficult and nebulous as meditation.*

54:50

*It's very difficult. Um, yeah, I hope that this feels like a space that's A little more open you know if you get distracted by something if you*

55:10

*get bored you start looking at Instagram on your phone that's fine*

55:30

*I was thinking about Laurie Anderson and her ability to just put things into words that feel very much like the very human base sort of niggling emotions that we all at some time or other feel feel like tapping into the body especially then this idea of like a nerve so you know I'm holding my nerve by doing this radio show and kind of inviting you all do the same doing this thing that feels can feel so alien this sort of occupying a different part of our brain where we do nothing and*

56:59

*explore what happens there together. But yeah, I spent when my last show was broadcast, I spent the hour feeling acute embarrassment that I was occupying the airwaves with dead air Oh, but that's just that's always going to be there, isn't it? Because this is a weird idea. And blah. Yeah. So yeah, this this, this feeling of the nerve Bible[7], like, was so. So connected to our nervous system. Like experiences of the world. Our responses to the world are all dictated by our nervous system. And I feel like switching off that part of my brain that's thinking what next? What next? What next? What next? allows my nervous system maybe to relax a little bit and take the foot off the gas and respond to self instead of the urgencies? I don't know. Yeah, thanks for staying tuned*

### 7. Losing and Re-Finding the Self, or Enacting and Enduring the Practice of Researching

In *Autotheory Theory*, Robyn Wiegman points to the moments within text when we speak without speaking. Language is not 'agent-less', it is not 'impersonal'. Spivak analyses a conversation between Foucault and Deleuze[8] to challenge their supposed attempt at theoretical production by identifying the text between the text, the 'track of [their] ideology' they unwittingly revealed. "Spivak's analysis punctures the 'much-publicised critique of the sovereign subject' by demonstrating how it 'actually inaugurates a Subject'" (Wiegman 2020).

The practice of autotheory punctures and re-punctures these lofty thought experiments. In my practice I attempt to understand and inhabit the malleability of the language around sovereignty, individual self-governance within a wider community, and (as I experience it) the soup of theory which surrounds these terms and their definitions. By occupying a radio space with my own thoughts, my own neurodivergence, I speak through my learning process, but I also begin to tease out what is unsaid, what is psychologically and philosophically unspoken, but still there.

The radio show has progressed through a series of experiments. Each month the show has been responsive to what I have been reading, listening to, unpacking. I read sections of text, invite audio clips from friends and collaborators, all constelling round this experiment with silence and waking sleep at the centre of the show.

As I navigate my struggles with independent study, the radio show has evolved, dictated by my boredom threshold, and my current interests. At times, when particular anxious about workload, the show has become a space to attempt reading as an endurance performance. The silence of a closed book can be deafening, the knowledge sealed tightly within its pages as I struggle to begin. Carving out this space to be and to begin has been vital.

As I continued to perform my relationship to research, to reading, I came across Adrian Piper's *Food for the Spirit*, described here by Fournier:

> Piper was told by her peers that she should read Kant, because his philosophical writings engaged with similar ontological and epistemological questions to those that Piper was engaging with in her studio work. She decided to start reading Kant, but rather than keeping this act separate from her studio practice, she framed it as part of a conceptual artwork. The performance, [ . . . ] consisted of her moving between monastically reading Kant's *Critique of Pure Reason* [ . . . ], scribbling in the margins as she read passages over and over, and capturing her

image on a photographic film negative using a Kodak Brownie camera and a mirror. (Fournier 2021a, pp. 71–72)

As female[9] researchers, the litany of (white, male) theorists we are told we should read becomes a groaning task we must incorporate. As much as we desire to be part of the theoretical trajectory, there is a reticence to add reverence to this over-exposed canon. I am advised that even if I am ambivalent to the canon, I must explain *why* I am ambivalent. Their contribution cannot go unacknowledged. Like Piper, I am drawn to performing this relationship, to bringing their texts inside the artistic practice. In *Food for the Spirit* the photographs act as proof, as audience, as a grounding mechanism, as an anchor for Piper's sense of self when this work on *Being* begins to feel too separate from being.

In a short text on this artwork, Laura Larson reflects on what is illuminated in this image (Figure 8), both visually and conceptually. The features of the room and the self-portrait "emerge and recede":

> The silence of photography becomes an apt and resonant space to stage the dissociative conditions of black life that insist on both visibility and the threat of disappearance.
>
> the fear of losing myself
>
> She is looking at herself and she's looking at us. She is a body and a spirit and she's an image. She is an image which is not a transcendent self. I imagine the time between the split seconds of these photographs and the swell of her consciousness. I imagine her interior life that can't be photographed. (Larson 2020)

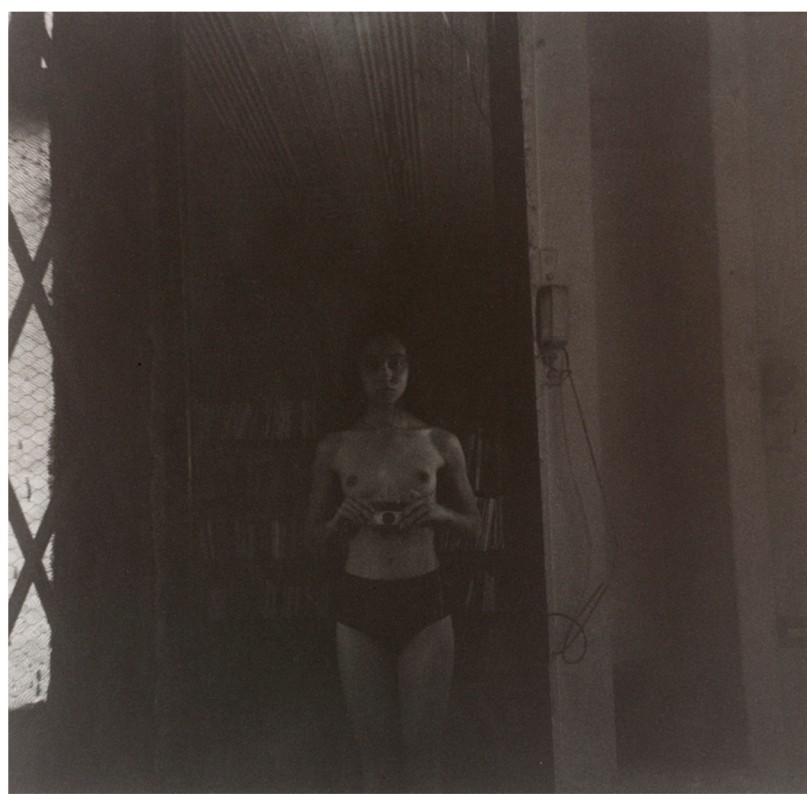

**Figure 8.** Adrian Piper, *Food for the Spirit*, 1971, Selenium toned gelatin silver print, 14 1/2 × 14 3/4 Courtesy of the RISD Museum, Providence, RI.

Autotheory both reveals, and cannot reveal, this internal space. When listening to an episode of *Holding My Nerve* the audience is in bed with me, on a picnic blanket with me, in the bath with me. They are in the most intimate relation to my body, while being entirely physically distanced. When working autotheoretically, we risk misinterpretation,

we risk the presumption of intimacy, we risk projection onto and mythology of the self. Performance art is so often conducted in silence, and while silent, we risk that others may read their own voices and interpretations into the empty space.

As part of a visiting artist lecture, Roy Claire Potter said "the problem of the threshold once again rears its head." (Potter 2021). This phrase has accompanied my work ever since, it articulates the negotiational, notional, nominal moves we make to exist and understand ourselves and the world around us. The threshold between the well and the unwell; the diagnosed and the undiagnosed; the self and the other; and the often-chasmic, sometimes-slight shift between silence and speech, between unread and read.

### 8. Extract of Episode 13 of *Holding My Nerve* (Figure 9)

Transcribed by Otter.ai and partially edited to amend inaccuracies, acknowledge silence and describe incidental sound
Grace Denton and Slacks Radio, June 2022

24:49
*to embrace where the mind goes and open up your thinking that way.*

24:58

25:19
*so I think my piece is becoming about the elusive nature of that so called scientific theory, I need to find what research has been done into it, which has been claimed in various spots on the internet that I found, but not actually referenced*

25:54
*cos I'm really interested in non, hmm, I'm not gonna say non medical approaches, but non medication? approaches?*

26:23
*I recently started taking some medication for ADHD which completely changed my life*

26:35
*felt like I was suddenly able to focus and get on with stuff that had been hanging around for years, and then I was told that it was making my heart rate too high, that I was technically tachycardic the entire time I was taking it [laughs awkwardly] so, I had to stop which is very disorienting*

27:11
*I am interested in these interventions and kind of brain hacks that do actually help if you can remember to do them*

27:47
*another aspect of this text I think is going to be about, about my kind of shame and self flagellation around not, not being able to just sit and*

28:14
*read about using this this format this radio format in order to help me perform the act of researching, the auto theoretical act of not being able to read the theory that you want to read, is that, [laughs] is that a thing?*

29:02
*so, with that I'm going to try and use this time to read. this odd context in the bath, with the recording set-up*



29:57
*so, today in this bath I'm gonna be reading the very recent text by Lauren Fournier about auto theory. I also notice that she refers back to Chris Kraus' I Love Dick which was a pretty key text for me in terms of accessing a [water rippling] critical, personal space. so I've got that here too. to refer back to. so I'm gonna leave you listening to the sounds, and I'll come back at the end to share any thoughts or quotes*

31:29
*[water dripping softly]*
*[pages turning]*

32:37
*[water splashes]*

33:50
*[page turns]*

34:02
*[water dripping]*

36:04
*[page turns]*

40:33
*[water rippling, bath squeaks]*

40:43
*so, in I Love Dick Kraus uses a crush that she has developed on a philosopher to fuel, a kind of re, recapturing of her own voice in relation to theory, I think the first half of the book or a large part of the book is written as a series of letters*

41:38
*to a semi mysterious figure called Dick, and in, in writing this text and revealing, revealing this kind of contradiction, of finding a kind of feminist voice but it being shamefully fuelled by a man who was by most accounts seemingly indifferent, she ostracised herself. so in 1997 when the book was first published, it was demonised, and then in 2016 it was it was published in the UK*

43:14
*and a whole, a whole new generation of people were wildly celebrating this, this text that Kraus was kind of over? [laughs]*

43:48
*this section that I'm reading and in the Fournier text is talking about how the celebration of the text in 2016 is perhaps to do with the more porous boundaries of privacy that we're accustomed to today. Kraus' radical blurring of art and life. So, one of the reasons why her work reverberates with millennial feminists in a post internet age of widespread disclosure*

44:35
*another, another byproduct of this kind of radical sharing is that, in reading, in rereading her work a lot of people from our generation were turned off by the fact that she talks about making her living from property in New York State and er,*

45:32
*I guess that's one of the, one of the risks of transparency and shifting attitudes to how we as artists make ends meet in this fucked up world*

46:48
*Interestingly as well the philosopher Dick Hebdidge, who the book was about but he was unnamed other than his first name, was only really widely associated with the book after he took legal action*

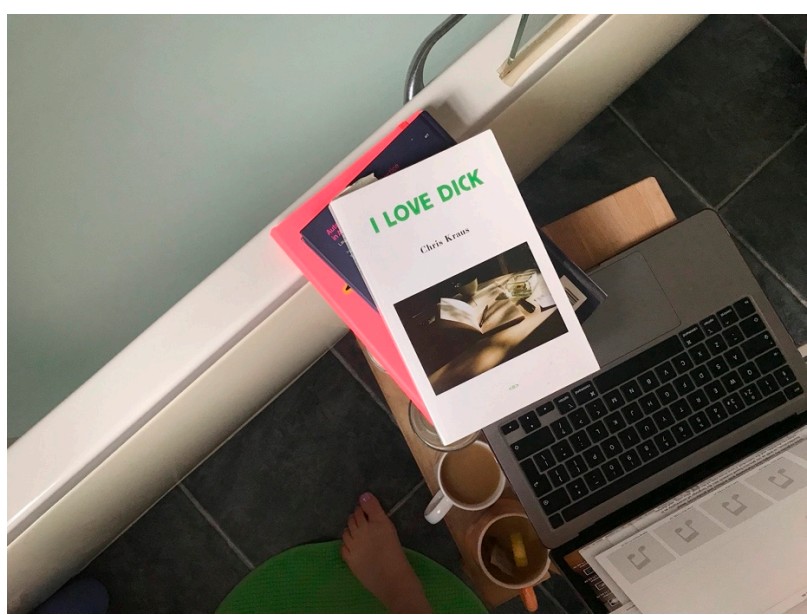

**Figure 9.** Episode 13 of *Holding My Nerve*, recorded in the bath while reading Fournier and Kraus Grace Denton and Slacks Radio, May 2022.

## 9. Conclusions

As I draw this article to a close, it now dawns on me that I have made this formerly ephemeral practice visible to the institute. This form of reflection feels necessary to the practice-based, autotheoretical process, but I am conflicted about engaging in a form of self-mythologising at odds with my own beliefs, my own aims. I have flagged my practice in a form palatable to the fabled 'REF'. This is the double-bind of the practice-based researcher. Our artworks are themselves research, we learn and uncover new knowledges by making, and to be explicit, by art-making. But in order for these practices to have weight, they require translation, validation, theoretical *under* pinning.

In literary autotheory, the theoretical is evident, it is cited, it is more often than not, readable. Moving forward, what are the practice-based ways I can *prove* the theoretical within my autotheoretical practice. It is a simultaneous, iterative, doing and undoing. Perhaps that is enough. Perhaps I can undo this analytical work by taking this piece back to the bath, back to the silence. Re-translate its tendrils and unmake these institutional trappings.

**Funding:** This research received no external funding.

**Data Availability Statement:** Not applicable.

**Acknowledgments:** I would like to thank the editors Cat Auburn, and Katherine Baxter for their trust, support and guidance, as well as my principal supervisor Christine Borland and my partner Craig Pollard, for listening and helping to join the dots.

**Conflicts of Interest:** The author declares no conflict of interest.

## Notes

1　Transcript of tape narration, Denton, Grace. Sixty Five. 2016. BALTIC 39, Newcastle upon Tyne.

2　A presentation format popular in the early 2010s in which 20 slides are visible only for 20 s each, moving the 'storyteller' on and keeping talks palatably brief.

3　At this point I must acknowledge the similarities between waking sleep and the practice of meditation. Thus far, my research has not progressed beyond a surface-level understanding of the deep rooted traditions of meditation, or Zen Buddhism, and this is a discrepancy I plan to rectify in further reflection, and writing on this project. In my introductions to Western translations of these Eastern principles, I have found a focus on self-discipline at odds with the neurodiverse mind. I am interested in this tension, and forging an accessible route through. The more I practice waking sleep, the closer I seem to come to the level of self-acceptance required to find solace therein. A succinct Buddhist parable no doubt exists to explain this process, but perhaps the internal barriers are greater in some than others, and we can convince ourselves we do not have the 'correct' capacity before we even try, making the external barriers that much harder to overcome.

4　There is a larger point to be made here, which is outside the scope of this essay, on Cage, Modernism and the obfuscation of the contribution of non-white artists and practitioners. I would point again to Craig Pollard's thesis on the Politics of Creative Practice. Several sections of his thesis were later published in his book *Inside A Gleaming Feeling* (2020), designed and printed by Glasgow's *The Grass is Green in the Fields For You*. The final section Kenny G and his Crew is of particular relevance here.

5　I believe the rise in the number of people seeking a diagnosis for ADHD is due to (1) a historic under-diagnosis of the inattentive form of the condition due to its characterisation within the popular imagination as the preserve of young, physically hyperactive boys, who struggle to sit still and make trouble with authority figures; (2) a rise in the distribution of information through easily digestible forms such as memes and short-form videos, for instance Tik Toks. This information boom and the subsequent rise in self-diagnosis is viewed warily by medical figures. Of course the 'infographic industrial complex' of social media does over-simplify complicated issues, and the desire for a quick fix for much deeper problems perhaps fuels this proliferation—nevertheless, this mode of communication is a successful tool for slipping in under the ADHD concentration limit, and meeting those suffering where they're already seeking dopamine.

6　Jo Hauge is a neurodivergent live artist doing a practice-based PhD about neurodivergent performance practice at Northumbria. They are based in Glasgow and are currently making work about fandom/figure skating/pleasure. Together we began a weekly coworking session for neurodivergent researcher-artists, called Making Time: http://www.gracedenton.co.uk/making-time (accessed on 21 June 2022).

7　Laurie Anderson's *Stories From the Nerve Bible*, a series of performances and then a book, 1972–1992.

8　In *Can The Subaltern Speak?* (1988) Spivak directly addresses a conversation between Michel Foucault and Giles Deleuze: From Nelson and Grossberg (1988, pp. 271–313).

9　I use the term female here as it relates directly to my personal experience, but I also use it in reference to Johanna Hedva's Sick Woman Theory, and their definition of 'woman' not necessarily as female, but as feminised, i.e "those who are faced with their vulnerability and unbearable fragility, every day, and so have to fight for their experience to be not only honored, but first made visible." (Hedva 2019, p. 8).

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
