# Peer review of "Holding Our Nerves—Experiments in Dispersed Collective Silence, Waking Sleep and Autotheoretical Confession"

_arts, 2022_

Round 1

Reviewer 1 Report

From Augustine to Teresa of Avila to Rousseau, autobiography has been a fraught, pathological necessity. With Barthes, then the internet, further layers of self-consciousness were introduced, though probably without altering the basic difficulty: to undress while remaining covered up, even if only by one’s skin. Author X as autotheorist is in a bind which, in a time of quantification, may be given the numerical value of 65. She decides, both in her radio show and the present paper, to perform her bind, in the first case as (intermittent) silence and in the second as verbal self-critique (to an extent a straightforward account of the bind but also a paper that acts out the bind). Performing the bind, on radio and in the paper, is painful (certainly) and (conceivably) liberating. Despite protestations, this is sometimes close to meditation, but a meditation which fails. But was meditation (or anything else) ever about success, i.e. with gain in mind? Or for its own sake? Anyway, the autotheorist knows hers is a test of endurance. The problem (of ADHD so-called) is understood as living out the problem. Is that its own solution? There’s no answer, just the bind reiterated. Is that enough (to keep going)? The autotheorist poses the question and has to make the question her answer. Ideally she would square the circle of knowing her being or being her knowing, on the face of it impossible—or perhaps not. She sees those contemporary dangers of selfie narcissism, self-indulgence, self-victimization, self-mythologizing etc. which come with performance. And the vicious (or non-vicious?) circle of pleading for and refusing the support of those other humans who may or may not be listening/reading. Well…acting out failure, freely choosing what can’t be helped, is a way out of sorts, or a way out when there’s no way out (Whitman: if I contradict myself, then I contradict myself). Might this strategy undo that fateful systemic 65? Possibly not (panic)…or possibly yes (the paper’s positive ending). The autotheorist’s choice, no one else’s.

The paper is recommended for publication.    

Author Response

Thank you for this poetic and expansive review, which provides ample provocation for future works, and responds, to the author's eye, in playful dialogue with the tone of the piece. 

Perhaps the cyclical nature of these ruminations is a feature of the autotheoretical, or the practice-based process - a repeated Sisyphean return to the question, the bind of 'having to make her question the answer'.

Despite pushing hard against the neoliberal urge to quantify and self-optimise, the perpetuation of the haunting number 65 has been maintained. Can this offering to the gods of academia finally purge the shame? Thank you for making it the autotheorist's choice.

Reviewer 2 Report

I was not familiar with the notion of autotheory before this request.  it seems to me that the paper responds precisely to the parameters and hopes laid out by the editors for this issue. My only reservation concerns the discussions of mindfulness, Zen Buddhism, and meditation.  The author(s?) misunderstand some of their principles, while eliding the three practices.  If anything they're perhaps describing orthodoxies associated with some Zen Buddhist meditation practices.  I urge them to look a little more deeply into what they're trying to use incorrectly as their counter example.

the footnotes might need copyediting (italicizing missing)

Author Response

I am grateful for the precise review, and the opportunity to amend based on the reviewer's feedback. Many thanks for the attentive reading.

My only reservation concerns the discussions of mindfulness, Zen Buddhism, and meditation.  The author(s?) misunderstand some of their principles, while eliding the three practices.  If anything they're perhaps describing orthodoxies associated with some Zen Buddhist meditation practices. I urge them to look a little more deeply into what they're trying to use incorrectly as their counter example.

I agree. I have added the following footnote to clarify my current position and future intentions, which will hopefully address concerns and put the elision into context. The footnote follows my first mention of meditation, on line 202:

"At this point I must acknowledge the similarities between waking sleep and the practice of meditation. Thus far, my research has not progressed beyond a surface-level understanding of the deep rooted traditions of meditation, or Zen Buddhism, and this is a discrepancy I plan to rectify in further reflection, and writing on this project. In my introductions to Western translations of these Eastern principles, I have found a focus on self-discipline at odds with the neurodiverse mind. I am interested in this tension, and forging an accessible route through. The more I practice waking sleep, the closer I seem to come to the level of self-acceptance required to find solace therein. A succinct Buddhist parable no doubt exists to explain this process, but perhaps the internal barriers are greater in some than others, and we can convince ourselves we do not have the ‘correct’ capacity before we even try, making the external barriers that much harder to overcome."

Reviewer 3 Report

I am intrigued by the author's radio show and the way that neurodivergence is addressed. An hour of nothing--how hard that is to do. The author/Grace Denton writes about their own work, but situates it in such a way that the reader of this paper can understand and follow the philosophical and feminist theoretical underpinnings of performing or embodying theory. The author cites the work of Laura Larson. Larson is primarily a photographer, whose work addresses many of the issues around embodiment and feminism in which the author is interested. I would suggest that the author look up Larson's work. 

Author Response

Many thanks for the incisive review. I am pleased to know that these reflections on my practice and their connections to other artists and theorists are doing the work I had hoped.

Thanks too for the prompt to look into Larson's work - how funny that a perfect reference artist can lay hidden in plain sight, within another reference.